# Exploiting Natural Niches with Neuroprotective Properties: A Comprehensive Review

**DOI:** 10.3390/nu16091298

**Published:** 2024-04-26

**Authors:** Hind Moukham, Alessia Lambiase, Giovanni Davide Barone, Farida Tripodi, Paola Coccetti

**Affiliations:** 1Department of Biotechnology and Biosciences, University of Milano-Bicocca, 20126 Milano, Italy; hind.moukham@unimib.it (H.M.); a.lambiase1@campus.unimib.it (A.L.); paola.coccetti@unimib.it (P.C.); 2National Biodiversity Future Center (NBFC), 90133 Palermo, Italy; 3Institute of Biology, University of Graz, Universitätsplatz 2, 8010 Graz, Austria; giovanni.barone@uni-graz.at

**Keywords:** neuroprotection, curcumin, resveratrol, ergothioneine, phycocyanin, Alzheimer’s disease (AD), Parkinson’s disease (PD), delivery strategies

## Abstract

Natural products from mushrooms, plants, microalgae, and cyanobacteria have been intensively explored and studied for their preventive or therapeutic potential. Among age-related pathologies, neurodegenerative diseases (such as Alzheimer’s and Parkinson’s diseases) represent a worldwide health and social problem. Since several pathological mechanisms are associated with neurodegeneration, promising strategies against neurodegenerative diseases are aimed to target multiple processes. These approaches usually avoid premature cell death and the loss of function of damaged neurons. This review focuses attention on the preventive and therapeutic potential of several compounds derived from natural sources, which could be exploited for their neuroprotective effect. Curcumin, resveratrol, ergothioneine, and phycocyanin are presented as examples of successful approaches, with a special focus on possible strategies to improve their delivery to the brain.

## 1. Introduction

Aging is a natural condition characterized by the progressive deterioration of physiological functions. It involves several different biological processes, and it is strongly influenced by environmental conditions, affecting lifespan and playing an essential role in the development of age-related diseases. Among these, Alzheimer’s disease (AD) and Parkinson’s disease (PD), involving cognitive and physical decline, are massive public health and social issues. From a molecular point of view, these diseases are characterized by neuroinflammation and the accumulation of protein aggregates. In detail, β-amyloid plaques and phospho-tau neurofibrillary tangles are found in the brain of AD patients, while α-synuclein fibrils are observed in PD patients. Despite the great efforts in developing new drugs against these pathologies, no effective or definitive treatment is currently available [1]. A complementary approach would consist of the prevention of these aging-associated diseases through the assumption of natural extracts or specific compounds derived from them, which can delay aging and prevent neurodegeneration.

The aim of this review is to present research developments in the use of molecules derived from natural sources (mushrooms, plants, microalgae, and cyanobacteria) with neuroprotective effects (Figure 1). In particular, the bioactivities associated with specific identified compounds are selectively taken into consideration in this manuscript. In fact, although several natural extracts have shown neuroprotective effects, for most of them, the specific mechanism of action is still not known.

Here, we describe compounds derived mainly from the secondary metabolic pathways of shikimate (such as phenols and flavonoids), terpenes, alkaloids, as well as polysaccharides. Furthermore, we discuss them as a function of their mechanisms of action through a narrative literature methodology. 

In addition, four widely studied molecules (curcumin, resveratrol, ergothioneine, and phycocyanin) are presented as model compounds for the development of novel neuroprotective molecules, with a focus on possible delivery strategies to improve bioavailability.

## 2. Active Natural Compounds and Targeted Bioactivities

A plethora of natural extracts derived from different matrices, such as plants, microalgae, cyanobacteria, or mushrooms, are rich in bioactivities. Searching scientific publications with “natural compounds AND neuroprotection” as keywords in bibliographic databases, 3864 results are shown via PubMed (https://pubmed.ncbi.nlm.nih.gov/?term=natural+compounds+AND+neuroprotection, accessed on 23 February 2024) and 11,060 through ScienceDirect (natural compounds AND neuroprotection—Search|ScienceDirect.com, accessed on 23 February 2024). In total, 2841 review articles and 3943 research articles (out of the 8558 manuscripts suggested by the ScienceDirect database) were published over the last 10 years (natural compounds AND neuroprotection—Search|ScienceDirect.com, accessed on 23 February 2024). This indicates a deep interest and prolific research activity in this field in recent years.

Neurodegenerative diseases, such as AD and PD, share some common features at the molecular level (e.g., protein aggregation, oxidative stress, neuroinflammation, and neuronal death) but also present disease-specific characteristics [2]. In the following section, we aim to examine the main processes involved in these pathologies and explore many emerging natural molecules presenting interesting bioactivities. Table 1 reports the classes of compounds and specific metabolites associated with their main bioactivities, which have been selected for this manuscript. Antioxidant properties were not extensively discussed since they have been widely explored in several other recent reviews [3,4,5]; therefore, they are only briefly mentioned here.

### 2.1. Anti-Amyloidogenic Compounds

Pathological protein aggregation is one of the hallmarks of neurodegenerative diseases and is associated with clinical outcomes since their presence often correlates with the progression of the symptoms [2]. However, mature aggregates or fibrils are not the major toxic species, while oligomeric assemblies are recognized as mainly responsible for cellular toxicity.

In AD patients, two distinctive aggregates are found that induce oxidative stress and neuronal loss. The first is represented by plaques of amyloid-β peptide (Aβ), a short proteolytic fragment derived from the amyloid protein precursor (APP) through the action of β- and γ-secretases. These insoluble deposits are found extracellularly near the synapses and lead to synaptic dysfunction and neuronal death. The second is represented by neurofibrillary tangles of the phosphorylated tau protein (p-tau), a microtubule-associated protein that, when hyperphosphorylated, accumulates intracellularly and leads to reduced neuronal functions and neurotoxicity. PD is instead characterized by the accumulation of aggregates of the presynaptic protein α-synuclein, which induces neurotoxicity and, finally, neuronal loss.

Therefore, compounds acting on protein aggregation have been widely studied in recent years (Table 1).

Different terpenoid molecules have been shown to affect protein aggregation, such as those from the black mushroom *Polyozellus multiplex* used in Asia (especially Korea, China, and Japan). Specifically, different p-terphenyls were purified from this mushroom, named polyozellin, thelephoric acid, and polyozellic acid, which strongly reduce extracellular Aβ1-42 release from the human Swedish mutant APP-transfected mouse neuroblastoma cell line [6]. Another interesting terpene derives from *Hericium erinaceus*, which is one of the most used medicinal mushrooms, also known as lion’s mane or Yamabushitake. The interest in this mushroom is due to the presence of a large amount of different bioactive metabolites, both in the fruit body and in the mycelia [81]. Among them, Erinacine A, a diterpenoid isolated from the mycelium, can reduce Aβ plaque formation in a mouse model of AD (APP/PS1 transgenic mouse), both by increasing Aβ degradation and inhibiting its production [7]. In addition, Erinacine A reduces ROS levels in mouse neuronal cells treated with a neurotoxin that inhibits complex I of the electron transport chain (ETC) and consequently also cell death [8]. 

Another intriguing bioactive terpenoid is Tanshinone IIA (Tan IIA), a diterpene phenanthrenequinone extracted from the roots and the rhizomes of the plant *Salvia miltiorrhiza*. Tan IIA presents a variety of bioactivities, including anti-apoptotic, anti-inflammatory, blood–brain barrier (BBB) protection, and antioxidant effects [82]. Tan IIA also inhibits amyloid fibril formation and disaggregates the Aβ1-42 amyloid peptide via specific binding to a hydrophobic β-sheet groove [10], as well as its reduced Aβ-induced toxicity in neuronal cells in vitro [9]. 

Also, phenolic compounds, like Oleuropein aglycone (OleA) and Hydroxytyrosol (HT), found in extra virgin olive oil, are known for their anti-amyloidogenic properties. These compounds exhibit strong antioxidant effects and are suggested to modulate the aggregation of the Aβ1-42 peptide, decreasing the formation of toxic aggregates [83,84,85]. OleA primarily interacts with the *N*-terminus of Aβ1-42, hindering the development of toxic oligomers and their transition into mature fibrils. In contrast, HT accelerates fibril formation, possibly due to increased hydrophilicity. Therefore, treatment with OleA and HT reduces the cytotoxicity of Aβ1-42, steering the aggregation process towards harmless outcomes even at low concentrations. Notably, evidence suggests that these polyphenols can penetrate the blood–brain barrier, accumulating in the brain parenchyma with regular intake [11] and supporting their development as an anti-aggregant in vivo.

Likewise, Salvianolic acid B (SalB), a dominant water-soluble component from *Salvia miltiorrhiza*, is known for its ability to ameliorate Aβ-induced neuronal degeneration and neuroinflammation in AD [12]. It has been reported that Salvianolic acid B not only inhibits Aβ fibril formation but also disaggregates preformed fibrils by converting them to an amorphous structure, thereby providing protection against Aβ-induced cytotoxicity. In a study in vitro, SalB was shown to significantly decrease the levels of Aβ1-42 in SH-SY5Y cells overexpressing the human APP (amyloid precursor protein) Swedish mutant [13]. 

Among the metabolites derived from the shikimic acid pathway, flavonoids represent a class of very interesting ones. Among these, Epigallocatechin gallate (EGCG), present in green tea, has received considerable attention in recent years due to its potential health advantages in preventing numerous chronic diseases, including neurodegenerative diseases. In transgenic mice overexpressing the mutant amyloid precursor protein (Tg APPsw), the administration of EGCG via intraperitoneal injection (20 mg/kg) led to a significant decrease in the brain levels of the Aβ peptide and a reduction in Aβ plaques. Additionally, EGCG promoted cleavage in the α-C-terminal fragment of APP, leading to elevated levels of soluble APP-α, indicative of enhanced α-secretase cleavage activity, which correlates with lower Aβ levels [14]. Strikingly, EGCG showed comparable effectiveness in diminishing Aβ deposition when administered orally via drinking water (50 mg/kg) [86]. In vitro, it was demonstrated that EGCG impedes the fibrillogenesis process of both α-synuclein and amyloid-β by directly interacting with the intrinsically disordered polypeptides, preventing their conversion into toxic species and promoting the formation of amorphous, non-toxic aggregates [15,16]. 

### 2.2. Inhibitors of AChE

Acetylcholine (ACh) is a neurotransmitter that activates cholinergic neurons in the central nervous system and is involved in the processes of memory and learning. Acetylcholinesterases (AChEs) are the enzymes responsible for the degradation of ACh and are potential targets for the treatment of AD since the dysfunction of cholinergic neurons is considered one of the major causes of cognitive impairment in AD patients. A second enzyme, butyrylcholinesterase (BChE), with a broader substrate specificity, seems to also be involved in ACh hydrolysis in the cholinergic synapse, and its levels are increased in the later stages of AD [87]. Therefore, a reduction in the level or the activity of AChE and BChE could lead to an increase in the level of ACh in the synaptic cleft, thus improving cognitive functions. Donepezil hydrochloride and rivastigmine are prescribed cholinesterase inhibitors that reversibly inhibit synaptic acetylcholinesterases, enhancing cholinergic transmission in AD patients [88,89]. However, these drugs present side peripheral cholinergic effects; therefore, the identification of new natural-based inhibitors of AChE activity is of valuable interest in the field. 

Several mushroom extracts have been shown to inhibit AchE activity, although in most cases, the effect was not attributed to specific compounds. *Ganoderma lucidum* is a Basidiomycetes fungus, highly used in traditional medicine in Asia since ancient times for its numerous health-promoting benefits [21]. In *G. lucidum*, several triterpenes were tested in vitro and showed AChE inhibition, with IC_50_ in the micromolar range [17]. Similarly, the above-mentioned p-terphenyls from the black mushroom *Polyozellus multiplex* present β-secretase non-competitive inhibitor activity in vitro [6].

Also, some alkaloids have been shown to inhibit acetylcholinesterases. Nostocarboline was isolated from the freshwater cyanobacterium *Nostoc 78-12A* and is a potent butyrylcholinesterase (BChE) inhibitor (IC_50_ of 13.2 μM) [18]. In cyanobacteria, the function of AChE inhibitors is required for the defense against the invertebrate larval settlement of competitor organisms [90,91]. Of interest is the anatoxin-a(s) compound, isolated from the cyanobacterium *Anabaena flos-aquae* strain NRC 525-17, which showed severe cholinergic overstimulation in rats along with a robust inhibition on both AChE and BChE enzymes in vitro [19]. 

Other alkaloids inhibiting AchE activity come from the Amaryllidaceae plant family. In fact, over 100 cultivars from the Amaryllidoideae subfamily have been studied for their AChE inhibitory activity [92]. Compounds derived from diverse skeleton types have demonstrated potent AChE inhibitory properties, thereby presenting avenues for structure optimization and the semi-synthesis of potent AChE inhibitors [93]. Among these, Galanthamine, which was approved in 2001 by the FDA for AD treatment, showed a strong AChE inhibitory activity (IC_50_ of 0.3–0.5 μM) [20], exerted by inhibiting substrate accommodation and hydrolysis in the active site of the enzyme [94].

### 2.3. Natural Compound with Anti-Apoptotic Effect

The most prominent effect of neurodegeneration is the resulting neuronal loss due to the ultimate activation of the apoptotic pathway. Apoptosis is crucial during the development of the central nervous system, but its improper activation can be detrimental in the adult brain and is involved not only in neurodegeneration but also in several other diseases [95]. 

Several compounds belonging to different chemical classes have been shown to prevent neuronal apoptosis in AD and PD models (Table 1). 

*G. lucidum* triterpenoids, which also present antioxidant activity, have been shown to decrease apoptosis both in vitro and in vivo in the hippocampus of an APP/PS1 mouse model of AD, with an effect on caspase-3 activation and on the ROCK signaling pathway [22]. This results in improved cognitive functions in the same transgenic AD mouse model, thus supporting the development of these triterpenoids as anti-apoptotic agents in neurons.

Another mushroom-derived terpenoid, the aforementioned Erinacine A from *H. erinaceus*, was shown to inhibit apoptosis induced by MPP (1-methyl-4-phenylpyridinium) treatment in a mouse neuronal cell line in vitro. This effect is associated with the increased phosphorylation of some proteins involved in survival, such as PAK1 and LIMK2 both in vitro and in the striatum of a 1-methyl-4-phenyl-1,2,3,6-tetrahydropyridine (MPTP)-mouse model of PD [8]. Another terpenoid regulating apoptosis is Astragaloside IV (AS-IVl), also known as Astragalus saponin IV, which is a primary bioactive component derived from the plant *Astragalus membranaceus*, which is widely used in treating various diseases in traditionally Chinese medicine. AS-IV can protect primary cerebral cortex neurons exposed to oxygen and glucose deprivation by modulating the PKA/CREB signaling pathway and preserving mitochondrial function [23]. AS-IV has also been shown to decrease the apoptosis rate in primary astrocytes, downregulating p-JNK, modifying the Bax/Bcl-2 ratio, and influencing caspase-3 activity [24].

Among the alkaloids, one of the most studied is caffeine (1,3,7-trimethylxanthine), a natural central nervous system (CNS) stimulant of the methylxanthine class. It is mostly sourced from coffee beans but can also be found in certain types of tea and cacao beans and is one common ingredient of many energy beverages [96]. It was shown that the PI3K/Akt signaling pathway, known for its role in cell survival, is implicated in the anti-apoptotic effects of caffeine by regulating the p-JNK and ERK protein kinase of both in vitro cultures [26] and in a mouse model of AD [25].

Fisetin (3,3′,4′,7-tetrahydroxyflavone) is another plant-derived compound that has garnered attention for its potential anti-apoptotic effects in the brain. It is a flavonoid abundant in fruit and vegetables, including apples, strawberries, cucumber, and persimmon [27,97]. In 6-hydroxydopamine-treated cells, which are used as a model of PD or traumatic brain injury (TBI), treatment with Fisetin inhibits the activation of key apoptotic proteins, including caspase-3 and caspase-9, suggesting a potential mechanism for preventing neuronal cell death [28,29]. Moreover, Fisetin has been observed to enhance Bcl-2 expression and reduce Bax expression in the mouse brain, thereby shifting the balance in favor of cell survival [30,31]. Furthermore, Fisetin treatment can reduce the expression of other apoptotic regulators, such as ASK-1, p-JNK, p53, and cytochrome C, in the cortex and hippocampus of mice [31]. The inhibition of these apoptotic molecules by Fisetin is triggered through the activation of the PI3K/Akt signaling pathway [28].

Finally, the flavonoid GardeninA (GA), found in the gum of the medicinal plant *Gardenia resinifera*, has shown a regulatory effect on apoptotic pathways in a *Drosophila* model of PD. Indeed, it was observed that GA pretreatment reduced the levels of active caspase-3 induced by paraquat (PQ), thereby mitigating apoptotic cell death. Moreover, GA pretreatment resulted in the downregulation of *reaper* expression, which was induced by PQ. This is significant as *reaper* can induce apoptosis by inactivating inhibitors of apoptosis proteins (IAPs), consequently allowing caspases to escape proteasomal degradation and trigger cell death [32].

### 2.4. Autophagy Stimulation

Autophagy is an evolutionarily conserved process that leads to the trafficking and degradation of intracellular components into the lysosome to maintain cellular homeostasis [98]. The targets of autophagy comprise misfolded or aggregated proteins, as well as damaged organelles, such as mitochondria, endoplasmic reticulum, and peroxisomes; in particular, mitophagy is the selective removal of damaged mitochondria via the macroautophagy pathway, which is crucial for monitoring their quality and decreasing cellular oxidative stress [99]. Autophagy is particularly important in neurons since it is involved in neurodevelopment as well as in the maintenance of neuronal homeostasis, and basal beneficial autophagy is necessary to maintain lifespan and prolong longevity [100]. Moreover, autophagy also exerts a key role in degrading aggregation-prone proteins, such as α-synuclein [101], β-amyloid [102], and the mutant protein Huntingtin [103]. For this reason, autophagy is considered a target for the prevention of neurodegenerative diseases, and a variety of natural compounds are reported to modulate autophagy and exert neuroprotective effects. Interestingly, most of the literature describes the efficacy of natural compounds in autophagy induction, and some of them also stimulate beneficial effects in particular conditions. Here, we collected the current evidence indicating their neuroprotective role (Table 1).

The aforementioned caffeine was shown to enhance autophagy by increasing the microtubule-associated protein, LC-3, reducing the receptor protein sequestosome 1 (SQSTM1/p62) and chaperone-mediated autophagy and regulating the expression of LAMP2A (lysosomal associated membrane protein 2). Chronic caffeine treatment re-establishes macroautophagy activity and broadly targets a variety of protein aggregates in the brain. Notably, caffeine treatment was shown to attenuate abnormal α-Syn aggregation and neurotoxicity by re-establishing autophagy activity in animal models of PD [33].

Most known plant-derived compounds regulating autophagy belong to the class of flavonoids (Table 1), such as Isoquercitrin, Hesperidin, Epigallocatechin gallate, Fisetin, and Morin.

Isoquercitrin, a flavonoid from *Apocynum venetum*, also known as quercetin-3-O-β-d-glucopyranoside, is a flavonoid that is also present in a variety of medicinal herbs, fruits, and vegetables as well as plant-based foods and drinks [104]. Isoquercetin can mitigate cellular damage in Schwann cells exposed to high glucose levels by upregulating the expression of Beclin-1 and LC3, thus reversing the inhibition of neural cell proliferation [34]. Likewise, in the model organism *C. elegans*, Isoquercetin counteracts the neurotoxic effects of peptide Aβ1-42 through the activation of both macroautophagy and proteasomal degradation pathways [35].

Hesperidin, identified as 3,5,7-trihydroxy flavanone-7-rhamnoglucoside, is a flavanone glycoside predominantly found in citrus fruits. In a study, Hesperidin treatment was able to counteract the increase in Beclin-1, LC3A, and LC3B induced by sodium fluoride, in this case, indicating its potential role in mitigating autophagy-related neurotoxicity. Furthermore, Hesperidin treatment was shown to stimulate the PI3K/Akt/mTOR signaling pathway, which is essential for cell survival [36].

The aforementioned Epigallocatechin gallate was shown to induce autophagy in SH-SY5Y cells by activating the Sirt1 pathway, specifically targeting the neurotoxicity and mitochondrial damage caused by the prion protein. EGCG induces an increase in LC3-II expression alongside a decrease in p62 levels, indicating enhanced autophagic activity [37]. Additionally, the natural flavonoids Fisetin and Morin have also been recognized for their ability to stimulate autophagy. Morin is highly present in the leaves, stems, branches, and fruits of plants belonging to the Moraceae family, and it possesses significant antioxidant and anti-inflammatory properties that offer neuroprotective benefits. In addition, Morin enhances mitophagy, which is essential for removing malfunctioning mitochondria. Strikingly, it accomplishes this by promoting the relocation of transcription factor EB (TFEB) to the nucleus and triggering the AMPK-ULK1 pathway. These actions have been proven to confer neuroprotection in a mouse model of PD induced by MPTP (1-methyl-4-phenyl-1,2,3,6-tetrahydropyridine), highlighting Morin’s potential [39]. As for Fisetin, it was observed that it can induce autophagy within cortical neurons by activating the Nrf2 transcription factors and TFEB, likely via the inhibition of mTORC1. Such activation is crucial for the disassembly of p-tau in cortical neurons, positioning Fisetin as a promising neuroprotective molecule [38].

### 2.5. Anti-Neuroinflammatory Compounds

Neuroinflammation is another typical hallmark of neurodegenerative diseases [2], which has been observed as an early event in AD development. Chronic inflammation involving microglia may occur in the presence of misfolded or aggregated proteins, high ROS levels, or damaged synapses. In addition, astrocytes are also activated in neurodegenerative diseases, affecting synaptic functions, neuron metabolism, protein aggregation, and, therefore, taking part in the disease process. For these reasons, the modulation of neuroinflammation is a relevant aspect of the prevention of neurodegeneration, and several compounds have been studied for their ability to modulate neuroinflammation (Table 1).

*Grifola frondosa*, also known as Maitake, is a well-studied medicinal mushroom used for centuries in China and Japan. It is particularly rich in polysaccharides, especially β-glucans, which have been shown to possess several health-promoting activities [105]. It was recently reported that treatment with proteo-β-glucans derived from Maitake increases astrocytes and microglia activation in a mouse model of AD and promotes the recruitment of microglia to the Aβ plaques in the cortex and hippocampal region, thus promoting the clearance of Aβ and reducing the Aβ plaque burden [40].

Another mushroom-derived compound, the well-studied terpenoid Erinacine A from the mushroom *H. erinaceus*, exerts an anti-inflammatory effect. Indeed, it reduces TNF-α expression upon lipopolysaccharide (LPS) stimulation in rat astrocytes and reduces NO production and iNOS expression in a mouse microglial cell line following treatments with LPS and IFN-α. Remarkably, Erinacine A treatment shows the same anti-inflammatory effect in vivo, leading to improved motor functions in an animal model of PD [41].

Considering triterpenoids, Astragaloside IV (AS-IV) also exhibits anti-inflammatory effects. In particular, it promotes the activation of nuclear factor-kappa B (NF-κB), reducing the expression of pro-inflammatory genes in microglial cell cultures and in vivo [42]. In addition, this molecule has been reported to modulate the activity of MAPKs and to inhibit the mRNA expression of various pro-inflammatory cytokines of cyclooxygenase-2 (COX-2) and inducible nitric oxide synthase (iNOS) enzymes, contributing to reducing the inflammation [42,43].

Among flavonoids, Fisetin exhibits a remarkable ability to decrease the expression of the glial fibrillary acidic protein (GFAP) in the cortex, hippocampus, and hypothalamus of mice, which is a hallmark of neuroinflammation [44].

GardeninA, among the class of flavonoids, reduces both iNOS and NO levels in the PD model in *Drosophila*, thus providing protection against NO-mediated inflammatory effects [32].

Finally, several lines of evidence indicate that caffeine can limit neuroinflammation associated with neurodegenerative disorders. Indeed, it was reported that caffeine reduces both microglia activation and astrocyte reactivity [49,51].

### 2.6. Compounds Improving Cognitive Functions

Neurodegenerative diseases are characterized by a strong and progressive cognitive decline. AD and PD patients have difficulties in learning and memory, as well as in speech, due to the neuroinflammation and progressive degeneration of neurons [106]. Thus, the evaluation of the effects of natural compounds on cognitive functions is often the biggest preclinical readout of the effect of these molecules. In Table 1, we report natural compounds that can improve cognitive functions; interestingly, some of them are also discussed as anti-neuroinflammation molecules (see the previous section). Indeed, as expected, the prevention of cognitive impairment is strictly connected to neuroinflammation reduction.

Different mushroom-derived compounds have been tested for their effects on cognitive functions. For instance, it was recently reported that treatment with proteo-β-glucans derived from Maitake mushrooms improves learning and memory dysfunctions in a mouse model of AD [40]. Another mushroom-derived compound, Erinacine A from *H. erinaceus* mycelium, was also shown to ameliorate behavior deficits in the same model, in particular, burrowing and nesting behavior, spatial learning, and memory [7]. Strikingly, in humans, two double-blind clinical trials showed that the *H. erinaceus* fruiting body supplement (about 3 g/day) improves cognitive functions in healthy Japanese 50-year-old individuals and individuals with mild cognitive impairment [53,54]. Although in these two studies no specific molecule was used, rather, the total dried fruiting body of *H. erinaceus* was used, and although it was not tested on patients affected by neurodegenerative diseases, these preliminary clinical trials suggest the potential of this mushroom to prevent cognitive impairment.

Among terpenes, Astragaloside IV treatment is able to reverse cognitive impairment, as indicated by improved learning and memory abilities in an AD mouse model [42], as well as in a rat model of vascular dementia [55].

Also, caffeine has been shown to potentially prevent memory decline in animal models of AD and PD. Moderate treatment with caffeine led to an enhanced capacity for memory and a reversal of AD pathology in mice [56], also preventing β-amyloid-induced cognitive impairment [57]. Additionally, caffeine consumption prevented hippocampal neurodegeneration induced by streptozotocin (STZ) in a rat model of sporadic dementia [58]. A study examining risk factors for AD in individuals over 65 years old found that coffee drinkers had a 31% lower risk of developing the disease [59]. Similar results were observed in a further study, which showed a reduced risk of AD/dementia in moderate coffee drinkers of 3–5 cups every day compared to those who consumed 0–2 cups of coffee over a 21-year period [60]. In a cross-sectional study involving 196 untreated patients with early-stage PD, coffee consumption was significantly associated with a reduction in the severity of the mood/cognition domain of the non-motor symptoms scale (NMSS) [61].

Numerous flavonoid compounds, including Morin, EGCG, Hesperidin, Isoquercitrin, and 6-Shogaol, have been documented to support the maintenance of cognitive abilities, as outlined in Table 1. Despite the high use of green tea polyphenols, there is a lack of satisfactory clinical trials involving EGCG in humans with neurodegenerative diseases, but several studies on animals are available. For instance, it was reported that the long-term oral consumption of EGCG (25 mg/kg) induced significant improvement in memory function in an AD mouse model [62]. In a study on cognitive performance, EGCG was found to enhance neurological effects during brain wave stimulation, resulting in a rise in tranquility and a reduction in stress [64]. Taken together, these data support the notion of EGCG dietary supplementation as a potentially safe and effective neuroprotective agent that should also be evaluated in subjects with neurodegeneration.

As concerns GardeninA, its ability to confer protection against mobility deficits induced by paraquat (PQ) in a *Drosophila* PD model has been examined, and it showed that pretreatment with GA significantly improved climbing performances [32].

Also, Morin was shown to improve learning and memory by attenuating oxidative stress, neuroinflammation, and neuronal death in the hippocampus in a rat model of AD [65]. Moreover, in an MPTP mouse model of PD, Morin (20 to 100 mg/kg) significantly reduced nigrostriatal lesions, dopaminergic neuronal death, striatal dopamine depletion, and long-term behavioral impairments [66]. Based on these studies, we can hypothesize that Morin could be beneficial in reducing cognitive dysfunction both in PD and AD.

Hesperidin has been demonstrated to facilitate the enhancement of spatial memory indirectly through the increased release of TGFβ-1 and the promotion of synaptogenic activity in cortical astrocytes of newborn Swiss mice [68]. Hesperidin addressed both motor and non-motor symptoms, such as anxiety-like behaviors and memory deficiencies, while concurrently mitigating monoaminergic abnormalities [107]. Another study investigated the impact of Hesperidin on learning and memory following prolonged exposure to mild stress. Albino Wistar rats, subjected to chronic mild stress and supplemented with Hesperidin (40 mg/kg for 28 days), demonstrated preserved cognitive functions compared to the control group. Notably, there was a statistically significant enhancement in memory consolidation observed after the learning process [69]. Strikingly, several clinical trials have also been conducted in humans, especially using Hesperidin-rich orange juice. In a clinical investigation involving 37 healthy adults aged 60–81, Hesperidin was tested at two different doses with the use of juice containing a high Hesperidin concentration (549 mg/L) or a lower Hesperidin content (64 mg/L). Following eight weeks, the group with elevated Hesperidin levels exhibited markedly enhanced cognitive function, executive function, and episodic memory compared to the group with lower Hesperidin levels [71]. A separate acute study involving 44 young adults consuming 500 mL of citrus juice with 42.15 mg of Hesperidin demonstrated a notable enhancement in cerebral blood flow (CBF) and executive function in the right frontal gyrus. However, no direct correlation was identified between behavioral improvement and the increase in CBF [70]. Another investigation centered on beverages rich in Hesperidin in healthy middle-aged men (30–65) revealed significant enhancements in attention, executive cognitive function, and psychomotor speed six hours after consuming orange juice with added orange pomace fiber (containing 220.46 mg Hesperidin) compared to a placebo group [67]. In a cohort study encompassing 13,373 Japanese elderly participants, the consumption of citrus displayed a dose-dependent and inversely correlated association with dementia incidence, indicating the potential safeguarding role of citrus flavonoids in diminishing the risk of dementia [72]. These studies proposed that dietary approaches rich in Hesperidin could potentially forestall cognitive decline in elderly individuals, although the specific mechanism remains somewhat unclear.

Among flavonoids, Isoquercetin has been attracting much attention for the treatment and prevention of AD because it contributes to the enhancement of cognitive performance, attenuating neurochemical and neurobehavioral changes in a rat model of AD-induced by colchicine through the activation of antioxidant and anti-inflammatory pathways [73].

6-Shogaol, Hydroxytyrosol (HT), Oleuropein aglycone (OleA), and Salvianolic acid B (SalB) are all recognized phenolic compounds that offer neuroprotection, particularly through the improvement of cognitive abilities. 6-Shogaol, a principal bioactive constituent of ginger, has demonstrated effectiveness in reducing cognitive impairments in animal models of dementia, highlighting its potential in combating cognitive decline. A study illustrated the influence of 6-Shogaol on memory deficits by affecting astrocyte and microglia activation in a mouse model of AD induced by injecting Aβ1-42 oligomers into the hippocampus. The results of the study revealed that 6-Shogaol treatment reduced inflammation and neuronal death, leading to an improvement in learning and mitigation of memory deficits [74]. In another investigation, significant enhancements were noted in behavioral and memory tests in a mouse model of AD after the oral administration of 6-Shogaol for two months. Consequently, the study proposed that 6-Shogaol, acting as an inhibitor of the cysteinyl leukotriene 1 receptor (CysLT1R) and cathepsin B (both implicated in the pathogenesis of AD), could act as a neuroprotective agent, preventing neuronal death and cognitive deficits [75]. Hydroxytyrosol (HT) and Oleuropein aglycone (OleA) are compounds found in extra virgin olive oil, and their dietary intake has been linked to protective effects against AD [108]. In an animal model of AD, the administration of HT (at 5 mg/kg/day for 6 months) slightly improved the cognitive behavior of mice [76]. Likewise, the administration of HT (50 mg/kg) to mice with AD for a duration of 8 weeks significantly improved their cognitive abilities, significantly reducing amyloid aggregates in the cortex [77]. Similarly, OleA is reported to protect against cognitive deterioration by reducing Aβ levels and plaque deposits in the animal brain together with increased synaptic function (LTP) [109]. Salvianolic acid B (SalB), extracted from the roots and rhizome of *Salviae miltiorrhiza*, is used in traditional Chinese medicine for its neuroprotective properties, attributed to its anti-inflammatory and antioxidant actions. As a water-soluble polyphenolic compound, SalB is characterized by low toxicity and high activity [110]. Notably, SalB demonstrated the ability to traverse the blood–brain barrier and directly impact the central nervous system. In a study involving 12-month-old male mice with cognitive impairment, treatment with Salvianolic acid B (20 or 40 mg/kg/day) led to improved cognitive function and the inhibition of neuroinflammation [78].

Besides SalB, *Salvia miltiorrhiza* contains Tanshinone IIA (Tan IIA), a lipophilic terpenoid compound with a pharmacological activity that enhances learning and memory in AD mouse models. Tan IIA treatments have exhibited preventive effects on behavioral tests in the early stages of the disease in six-month-old males. In the conducted research, which involved the Novel Object Recognition test, it was noted that Tan IIA was able to mitigate short-term memory deterioration in mice. Additionally, during the Morris Water Maze test, Tan IIA reversed the impairments and improved the swimming strategy with which the mice tracked down the platform, suggesting an attenuation of spatial memory deficits and an improvement in the learning capability of Tan IIA treatment [80].

## 3. Four Widely Explored Molecules with Neuroprotective Effects

As highlighted in the previous section, the potentialities of natural compounds are limitless, and an increasing number of manuscripts describing their neuroprotective effects have been published in recent years. In this section, we describe in more detail four examples of natural compounds that have been extensively studied as neuroprotective agents, having shown effects on multiple processes involved in neurodegeneration, and which are currently under investigation in clinical trials for their great potential (Figure 2).

### 3.1. Curcumin

Curcumin is a natural polyphenol derived from the rhizomes of the turmeric plant *Curcuma longa* [111], also called the “spice of life” for its history deeply interwoven with cultural, medicinal, and culinary traditions [112]. Curcumin is the primary component of turmeric, responsible for the plant’s intense yellow color [113], and has emerged as a focal point in extensive scientific research for its potential therapeutic application. In particular, scientific evidence from in vitro studies, in vivo experiments, and clinical trials highlights the neuroprotective effect of curcumin, which modulates several pathways associated with neurodegenerative diseases [114].

Like several polyphenol compounds, curcumin acts as a scavenger for free radicals and enhances antioxidant enzymes such as superoxide dismutase (SOD), catalase (CAT), and glutathione peroxidase (GPx), thus protecting neurons from oxidative stress [115].

Curcumin also has anti-inflammatory effects by inhibiting diverse inflammatory markers, including pro-inflammatory cytokines, TNF-α, IL-1β, and IL-6, thus mitigating the brain’s inflammatory response [116]. Additionally, it downregulates the activity of pro-inflammatory enzymes, such as COX-2 and iNOS, which lead to reduced levels of prostaglandins and nitric oxide [117]. It also modulates the signaling pathways of NF-kB, a key transcription factor involved in the regulation of inflammatory responses, along with Wnt5 (Wnt Family Member 5A) and JNK1 (c-Jun *N*-terminal kinases), which are both crucial for neuronal cell survival, inflammation, and apoptosis [118]. Interestingly, curcumin has been investigated for its ability to interfere with the misfolding and aggregation of proteins linked to neurodegenerative disorders, including Aβ1-42 and α-synuclein [119]. Based on experimental findings, curcumin has been shown to directly bind to misfolded proteins, altering their conformation and hindering their ability to aggregate into toxic forms [120,121]. In addition, it promotes the degradation of misfolding proteins through cellular protein degradation pathways, such as the ubiquitin–proteasome system and autophagy, helping to prevent the accumulation of toxic aggregates [122]. Indeed, many studies have demonstrated curcumin’s important role in inducing macroautophagy. Specifically, in PD models both in vitro and in vivo, curcumin activates autophagy through the inhibition of the Akt/mTOR pathway, contributing to the neuroprotective effects. Interestingly, it was found that curcumin supplementation substantially increases the LC3-II/LC3-I ratio and Beclin-1 and reduces the accumulation of α-synuclein [123,124]. In keeping with this, curcumin downregulates the PI3K/Akt/mTOR pathway and increases the expression of Beclin-1 and the LC3-II/LC3-I ratio, promoting the formation of autophagosomes and reducing the aggregation of Aβ in a mouse model of AD [125].

Several clinical trials have been performed to address the clinical potential of curcumin as a neuroprotective agent in humans. Initial placebo-controlled trials in subjects with mild-to-moderate AD did not show any evidence of efficacy [126,127], probably because of the limited bioavailability of curcumin used (despite the high dose of up to 4 g/day) or due to the short time of the trial (6 months). However, the use of a solid lipid formulation of curcumin (Longvida, about 80 mg/day of curcumin) in healthy elderly subjects suggested an increase in attention, as well as in working memory and mood upon acute administration (1 h) and after 4 or 12 weeks of administration [128,129]. Interestingly, the administration of a lipidated form of curcumin (80 mg/day) also increased catalase activities and decreased Aβ concentration in the plasma after 4 weeks of treatment [130]. In a longer (18 months) double-blind placebo-controlled trial using a bioavailable form of curcumin (Theracurmin, 180 mg/day), curcumin treatment led to significant memory and an improvement in attention in non-demented adults. Strikingly, these results were accompanied by a reduction in FDDNP-PET (2-(1-{6-[(2-[fluorine-18]fluoroethyl)(methyl)amino]-2-naphthyl}-ethylidene)malononitrile positron emission tomography) binding, which can indicate the brain deposition of amyloid plaques and tau tangles, particularly in the amygdala and hypothalamus, which are crucial regions for mood and memory [131]. All these findings support the use of curcumin to preserve cognitive functions in humans, although high attention must be paid to its bioavailability (see also Section 4). 

Nowadays, curcumin is widely used not only in the food industry but also in the pharmaceutical and cosmetic industries. The global Curcumin market is expected to undergo substantial growth from 2023 to 2030. The market size was USD 207 million in 2022 but is projected to increase to USD 265 million by 2028. India is at the forefront of this market, contributing over 70%, followed by China and the USA. Strikingly, pharmaceutical-grade Curcumin represents the largest product segment, accounting for over 65% [132].

### 3.2. Resveratrol

Resveratrol is a natural polyphenol present in wine, red grapes, berries, chocolate, and peanuts, and it is attracting more and more attention due to its potential neuroprotective properties [133]. In fact, numerous studies suggest that resveratrol may safeguard the brain through various mechanisms. The compound’s antioxidant properties are among the most studied activities of resveratrol, which is able to neutralize harmful free radicals that can damage brain cells. Resveratrol showed potent antioxidant activity in hippocampal neuronal cells by scavenging free radicals and shielding against NO toxicity [134]. Furthermore, it reduces quinone reductase 2 activity, which contributes to age-related metabolic stress and cognitive deficit in hippocampal neuronal cells treated with menadione [135], and also upregulated the endogenous antioxidant response in a mouse model exposed to ischemia [136]. Resveratrol also activates protein kinase AMPK and its upstream regulator, LKB1, in neuronal cell lines, primary neurons, and the brain [137]. In addition, it activates the Nrf2 pathway, regulating antioxidant enzyme expression in a number of in vitro models (PC12 cells, astrocytes, cerebellar granule neurons, and oligodendroglial cells) as well as in in vivo models [138]. In primary microglia cultures exposed to LPS in rats, resveratrol decreases the production of prostaglandins, NO, TNFα [139], and COX-1 [140], showing strong anti-inflammatory properties, also reported by other studies [140,141]. Resveratrol activates SIRT1, leading to a reduction in the NF-κB signaling pathway in glial cells exposed to Aβ1-42-induced toxicity. This effect is particularly relevant since the down-regulation of SIRT1 expression abolishes the protective action of resveratrol in neuroblastoma cell lines exposed to the neurotoxic compound 6-hydroxydopamine (6-OHDA) [142]. 

Interestingly, resveratrol binds to the peptide Aβ1-42 in vitro, and by changing the conformation of oligomers, it probably attenuates their cytotoxicity [143]. 

The neuroprotective effects of resveratrol have also been tested in human subjects in randomized, double-blind, placebo-controlled clinical trials. In healthy young men, a single oral dose increased cerebral blood flow in a dose-dependent manner, but no effect on cognitive function was observed [144]. However, another trial on young adults receiving resveratrol for 28 days demonstrated cognitive benefits related to working memory tasks [145]. Resveratrol (150–200 mg/day) has also been tested in elderly subjects. Initial evidence suggested improved mood and cognitive performances in post-menopausal women after 14 weeks of administration [146] and improved retention of words over a 30 min delay after 6 months of administration [147]. However, the following study failed to show any effect of resveratrol on verbal memory performance [148]. In a longer study (52 weeks) on AD patients, a higher dose of resveratrol (up to 2 g/day) decreased the level of metalloproteinase MMP9, which is involved in maintaining the integrity of the BBB, in the cerebrospinal fluid, and slowed down progressive cognitive decline [149], strongly supporting the role of resveratrol as a neuroprotective agent also in AD patients.

Despite challenges such as low bioavailability, resveratrol tolerability, and its safety profile position it as a promising candidate for long-term treatment against cognitive decline in humans [150], and advanced pharmaceutical technologies aim to improve resveratrol’s bioavailability [151]. 

The increasing demand for resveratrol is attributed to its widespread use across various sectors, including cosmetics, dietary supplements, medications, and personal care items. Projections for the global resveratrol market indicate substantial growth between 2023 and 2030. As of 2022, the market is steadily expanding, and with significant strategies adopted by key players, it is expected to surpass the anticipated growth trajectory. As of 2021, the global resveratrol market size was estimated to be USD 165.73 million. Forecasts suggest a Compound Annual Growth Rate (CAGR) of 8.52%, leading to a market value of USD 270.69 million by 2027. Leading the resveratrol market are North America, Mexico, and South America, Asiatic-Pacific areas, the Middle East, Africa, and Europe [152].

### 3.3. Ergothioneine 

Ergothioneine (EGT) is a non-proteinogenic, sulfur-containing amino acid characterized by good stability and solubility. Its name comes from the ergot fungus *Claviceps purpurea*, from which it was first isolated in 1909. It is an essential amino acid that can be introduced into the diet and is particularly abundant in mushrooms, although it can also be found in certain meat products (such as chicken liver) and grains and at a much lower concentration in other food sources, like eggs, rice, garlic, and beans [153]. It is absorbed in the gut through the transporter OCTN1 (the organic cation transporter novel, type 1) [154] and is also highly distributed in the brain. Importantly, in the brain, OCTN1 is expressed in neurons, dendrites, and microglia but not in astrocytes [155,156]. Strikingly, a decline in EGT has been observed in patients with dementia, including AD [157], suggesting that EGT reduction may be associated with a higher risk of brain diseases. Thus, recently, EGT has gained great attention as an anti-aging compound, and it is often called the “longevity vitamin” [158,159,160]. The role of EGT as an antioxidant agent has been widely investigated; it acts both as a direct reactive oxygen species (ROS) scavenger as well as through the activation of the antioxidant cellular processes mediated by Nrf2 [153]. It was also observed that EGT reduces Aβ plaques in a 5XFAD mouse model of AD [161].

Despite these data, the effect of EGT administration on human subjects was poorly explored. In 2017, EGT uptake and clearance were measured, but very limited effects on biomarkers of oxidative damage were observed, probably due to the short time of administration (7 days) [162]. Some other clinical trials are ongoing (see www.clinicaltrials.gov, accessed on 15 February 2024), but at the moment, no clear results on humans have been reported, and further studies are required to clarify EGT health benefits in humans. 

Yet, EGT is widely commercialized as a diet supplement for its antioxidant and anti-aging properties, especially in the United States, China, and Europe. The price of EGT is about USD 700/kg; the global market of EGT was estimated to be USD 15 million in 2020 and is expanding more and more; it is expected to reach USD 125 million in 2027 [163], thus representing a huge developing market. For these reasons, new strategies for EGT extraction from mushrooms, but also for production from engineered strains and through chemical synthesis, are being developed [163].

### 3.4. Phycocyanin and Spirulina Extract

Phycocyanin (PC) is a pigment–protein complex from the light-harvesting phycobiliprotein family, which also includes Allophycocyanin, Phycoerythrin, and Phycoerythrocyanin [164,165]. Phycobiliproteins aggregate to form clusters that adhere to the membrane, called Phycobilisomes. PCs are found in cyanobacteria as an accessory pigment to chlorophyll and with a molecular weight of around 30 KDa. PCs are composed of two subunits, α and β, combined together [166,167,168]. These proteins have a light blue color, absorbing orange, and red light near 620 nm and emitting fluorescence at around 650 nm. The light absorption and the fluorescence emission are type-dependent (e.g., Allophycocyanin absorbs and emits at longer wavelengths than C-Phycocyanin or R-Phycocyanin). Approximately 20–45% of the protein fraction from the cyanobacterium *Arthrospira platensis* (Spirulina) is PCs [169,170]. The utilization of an appropriate extraction method is of high importance to obtain a pure and stable PC since the release of this pigment depends on the cell membrane disintegration [171].

Several studies describe PC’s beneficial characteristics and antioxidant properties in vitro and in vivo [165,172]. In particular, PCs are able to scavenge reactive oxygen and nitrogen species to counteract lipid peroxidation and to inhibit enzymes, such as NADPH oxidase and COX-2 [173]. In animal models of multiple sclerosis and ischemic stroke, these compounds induce remyelination, as demonstrated by electron microscopy, also showing a reduction in pro-inflammatory cytokines and induction of immune suppressive genes [173]. Several studies focus on the effects of Spirulina in the brain, highlighting its beneficial anti-inflammatory and antioxidant effects acting on glial cell activation and in the prevention and/or progression of neurodegenerative diseases (in particular, AD and PD) [174].

Spirulina is not only rich in PC but is a rich source of other bioactive compounds, including proteins, polysaccharides, polyunsaturated fatty acids, vitamins, and carotenoids [175,176,177]. Different studies have demonstrated the neuroprotective role of Spirulina on the development of the neural system, senility, and a number of pathological conditions, including neurological and neurodegenerative diseases [178,179,180,181].

Extracts of *Spirulina maxima* (70% ethanol) were studied in a 12-week double-blind placebo-controlled clinical trial in the elderly, showing a significant improvement in visual learning and visual working memory [182], although the exact components of this extract were not specified by the authors. 

Nowadays, cyanobacteria- and microalgae-based bioactive compounds are generally commercially available. Their market demand will continue to grow due to the increasing demand for natural products in the food industry [177]. The global market regarding microalgal-based proteins is foreseen to reach USD 0.84 billion by 2023 [183]. In particular, C-Phycocyanin is expected to reach a market value of USD 409.8 million within the 2030s [183]. Spirulina, as a protein-rich cyanobacterium (60–70% (*w*/*w*)) and an important source for C-Phycocyanin, has also gained attention as a promising feedstock for large-scale production. It is cultivated in different countries (e.g., Germany, Portugal, the USA, and China) and mainly commercialized as dried biomass. The exclusive production of high-value proteins and C-Phycocyanin is still emerging.

## 4. Delivery Strategies

We have described how a plethora of natural compounds exhibit beneficial activities [184]; nevertheless, the translation of the huge number of preclinical studies to clinical trials is, of course. very challenging. One of the biggest issues is represented by the fact that most bioactive substances are environmentally sensitive, with low solubility, chemical instability, and low bioavailability, as well as an inappropriate molecular size [185]. To overcome these problems, diverse strategies have been employed to improve the bioavailability of natural products [186]. Recently, there has been an increasing interest in nanotechnology, which encompasses the generation and utilization of materials within the nanometer scale [187] to deliver natural compounds, leading to numerous advantages. Nanoparticles are favored for their abundance, biocompatibility, bioavailability, biodegradability, increased chemical and physical stability, as well as for extended shelf-life of nanoparticles [188]. Nanoencapsulation is a new frontier in nanoscience in the food, pharmaceutical, and cosmeceutical industries. This technique consists of enclosing a bioactive compound or a drug in liquid, solid, or gaseous states within a matrix or inert material, commonly using a polymer matrix or ‘wall material’. This encapsulation aids in enhancing protection, increasing stability, solubility, and availability through the cells [189]. The choice of the preparation method for an agent’s encapsulation depends mostly on its properties, such as the state of aggregation, sensitivity, and especially the size of molecules. In the last few years, numerous methods for the encapsulation of natural products, such as extract or pure natural compounds, have been developed. 

Some of the compounds described in the previous section have been nano-formulated to improve their functional properties. Among them, curcumin (described in Section 3.1), due to its low solubility, biochemical degradation, and poor bioavailability, has been encapsulated in nanoemulsion, which has been described as an excellent carrier of lipophilic curcumin [190,191]. In nanoemulsion delivery systems, curcumin molecules are typically located within the hydrophobic interior of oil droplets. In this way, curcumin is protected from active substances within the aqueous phase that would typically promote its chemical degradation. In addition, this formulation leads to enhanced intestinal absorption of curcumin, which is related to the inhibition of curcumin metabolism by nanoparticles based on hydrogen bonding-driven self-assembly. The enhanced cellular uptake of curcumin by encapsulation also provides pH-triggered intestinal targeted release properties and strong antioxidant capacity [192].

Curcumin’s stability is affected by the composition and structure of delivery systems. Numerous studies have demonstrated that the stability of Curcumin is influenced by the nature and concentrations of emulsifiers. Kharat and colleagues examined the impact of emulsifier type (sodium caseinate, Tween20, quillaja saponin, gum arabic) and concentration on the stability of curcumin-loaded nanoemulsions. They found that the degradation of Curcumin in saponin-stabilized nanoemulsions accelerated curcumin degradation (after 15 days) due to its ability to promote peroxidation reactions [193]. Similarly, Artiga-Artigas et al. assessed the impact of three molecularly different surfactants, lecithin, Tween20, and sucrose monopalmitate, and their concentration on the stability of Curcumin-loaded nanoemulsions and evaluated their antioxidant capacity. They found that Curcumin-loaded nanoemulsions with lecithin showed long-term stability and exhibited higher antioxidant capacity compared to those made with the other surfactants. This enhanced performance is attributed to lecithin’s phosphate ions, which can form hydrogen bonds with phenolic hydroxyls of curcumin [194].

Many studies have shown that the bioavailability of curcumin within the gastrointestinal tract (GIT) can be greatly increased by emulsion-based systems compared to crystalline curcumin dispersed within water [195,196]. Researchers have subjected curcumin formulations to a simulated GIT model consisting of the mouth, stomach, and small intestine phases. They have shown that the bioaccessibility value of curcumin encapsulated (74–79%) is significantly higher than that of the curcumin solution (10%). In addition, it suggests that encapsulating curcumin within small lipid particles may be advantageous for improving its absorption from the GIT [197].

With a similar approach, polysaccharides from *G. lucidum* were also encapsulated in a nanoparticle to achieve better health-promoting effects. However, *G. lucidum* polysaccharides have some challenging characteristics, such as low solubility and low yield. Accordingly, to surmount these problems, a polysaccharide-based nanoparticulate co-delivery system was developed to obtain excellent functional properties and processing benefits [198]. Correspondingly, research demonstrated that *G. lucidum* polysaccharides in nanoparticles not only perform their original functions but also contribute to preventing the rapid metabolism of other active components. This leads to an improved loading rate and encapsulation efficiency, thereby enhancing the bioactivity and pharmacological effects of polysaccharide-based nano-systems [199]. 

As mentioned above, Spirulina provides a high protein content and a full range of essential amino acids, which is why it has become of special interest as a good protein source. Additionally, it exhibits antioxidant and anti-inflammatory effects on the human body, as presented in Section 3.4. Nevertheless, it brings some challenging characteristics that limit its use in food products, such as the strong, dark, green-blue color, derived from pigments of chlorophyll A, phycocyanin, and carotenes. Therefore, nanotechnology, such as microencapsulation by spray drying and enzymatic treatment, has become of interest to overcome Spirulina’s restricted use [200]. Maag et al. applied microencapsulation and additional enzymatic hydrolysis to the Spirulina extract in order to reduce the intense green color, optimize thermal stability, increase bioavailability, enhance protein powder solubility, and moreover, improve digestibility. This method consists of entrapping sensitive nutrients within a wall material maltodextrin and gum arabic in order to protect them from physico–chemical influences and external factors, such as heat and oxidative processes caused by air and light. They have shown that enzymatic treatment or encapsulation enhances the protein solubility between 10 and 30% compared with the untreated samples. Furthermore, a considerable reduction in green color intensity was observed in the encapsulated samples [201].

When considering neuroprotective agents, the BBB is a major challenge against obtaining the optimum effect from natural compounds [202]. Also, in this case, nanotechnologies, by means of various drug delivery systems, propose good strategies owing to their unique ability to target the BBB and to assist in the better delivery of natural compounds to the brain than natural compounds on their own. In addition, with the help of the surface modification of nanocarriers to take advantage of receptors overexpressed at the BBB, nanocarriers have been shown to efficiently overcome the BBB and deliver the selected compound to specific sites within the brain. To exemplify, the surface modification of curcumin nanocarriers incorporates targeting ligands to bind receptors and transporters at the BBB, which enables them to bypass receptor-mediated endocytosis, thus improving central nervous system selectivity and permeability [203]. Another study reported that curcumin-encapsulated poly(lactide-co-glycolide) nanoparticles (cur-NPs) induced the proliferation of neural stem cells and their differentiation to neurons in vitro and in the hippocampus and subventricular zone (SVZ) of adult rats. The nanoparticles increased curcumin levels in the rat brain compared to similar doses of bulk curcumin, and the presence of cur-NPs in the hippocampus and SVZ confirmed the ability of these nanoparticles to cross the BBB and induce neurogenesis [204].

Bidirectional communication between the human gut microbiota and the brain, commonly referred to as the microbiota–gut–brain axis or gut–brain axis, exerts a significant influence on behavior and brain function. The microbiota–gut–brain axis represents an intricate communication network that connects the gastrointestinal tract with the brain and facilitates bidirectional interactions through neural, endocrine, and immune pathways [205]. The investigation into dietary impact on the gut microbiota demonstrates a major impact on the brain, especially on the development and function of microglia and the BBB [206]. The gut microbiota produces various neurotransmitters, such as GABA, dopamine, and serotonin; amino acids as well as their derivatives, such as tryptophan and tyramine; and microbial metabolites, like short-chain fatty acids and aryl hydrocarbon receptor (AhR) ligands. These compounds can influence the local gut physiology and can circulate through the blood to interact with the host immune system, which directly signals the brain and may affect microglial activation [207]. Notably, early microbiome changes were found in patients with preclinical AD and PD, and compelling evidence shows that the altered gut microbiome can drive neurodegenerative disease pathogenesis [205]. Strikingly, changes to the gut microbiome can cause the misfolding and abnormal aggregation of α synuclein, which can be transported from the periphery to the central nervous system (CNS) via the vagus nerve [208].

This insight into the microbiota–gut–brain axis makes it a promising target for neuroprotective strategies. In fact, adjusting the gut microbiota through dietary changes could potentially influence microglial activity and mitigate the progression of neurodegenerative diseases [205]. Therefore, the bioavailability of natural compound nanoparticles within the gastrointestinal tract can affect the gut microbiota, thereby impacting brain-related diseases. For instance, Qu and colleagues developed Honokiol nanoparticles, a plant bioactive compound, which demonstrated the improved bioavailability and bioactivity of Honokiol. It has been shown that this nanoparticle suppresses the neuroinflammatory response, regulating the gut microbiota, reducing β-amyloidosis, and, consequently, exhibiting neuroprotective properties in a mouse model of AD [209].

To summarize, the use of nanoparticles is a promising approach to deliver natural compounds due to their abundance, biocompatibility, bioavailability, biodegradability, and increased chemical, physical, and shelf-life. The bioavailability of natural compound nanoparticles affects the gut–brain connection, thereby preventing neurodegenerative diseases. Nevertheless, even if it is successful in various in vitro and in vivo BBB models, further studies are needed to investigate the efficacy of these approaches in humans. In addition, despite the success of this technique, it is imperative to evaluate more types of nanomaterials and identify additional potential health effects.

## 5. How a Better Understanding of Natural Molecules with Neuroprotective Effects Could Improve Their Utilization in the Next Decades

A better understanding of natural molecules with neuroprotective effects could lead to the development of more effective strategies to prevent neurodegenerative disorders [210]. The free radical theory of aging postulated by Harman in 1956 points out that excessive production and accumulation of ROS causes a subsequently altered cellular integrity that eventually leads to severe and irreversible damage [210,211]. Therefore, oxidative stress plays a crucial role in neuronal damage associated with aging as well as neurodegenerative disorders [210,212,213]. Epidemiological and biochemical studies have identified food components as promising agents for neuroprotection [210]. Neuroprotection includes mechanisms such as the activation of specific receptors, changes in enzymatic neuronal activity, and the synthesis and secretion of different bioactive molecules. All these mechanisms are focused on preventing neuronal damage and alleviating the consequences of massive cell loss. Some neuropathological disorders selectively affect particular neuronal populations, but their neurochemical and anatomical properties in order to design effective therapies should be known. Although the design of such treatments could be specific to neuronal groups sensitive to damage, the effect would have an impact on the whole nervous system [210].

The development of new strategies, like nanocarriers (see Section 4), that help to promote the access of neuroprotective molecules to the brain is needed to provide more effective therapies for disorders of the central nervous system (CNS) [210]. In order to trace the success of these nanoplatforms on the release of the bioactive cargo in the CNS and determine the concentration at trace levels of target biomolecules, the use of analytical chemistry and concrete separation instrumental techniques are essential. Currently, these techniques are used for the determination and identification of natural neuroprotective molecules in complex matrixes at different concentration levels [210].

An alternative, not a mutually exclusive approach, is represented by the chemical modification of natural compounds to mainly increase their stability and/or solubility. Researchers have chemically modified several natural compounds with neuroprotective or anti-aging activity [213,214,215]. Fucoidan is a class of sulfated polysaccharides rich in fucose, and fucoidans from different species have different structures [216]. As investigated via electron microscopic images and signaling pathway studies, this compound, despite the fact that does not inhibit the aggregation of Aβ1-42, can block caspase-3 and caspase-9 and then apoptosis [217]. Naive Bayesian and recursive partitioning algorithms were applied by Fang et al., for example, to construct classifiers to predict active molecules against 25 key targets toward AD using the multitarget-quantitative structure–activity relationships (mt-QSAR) method [218]. In this context, fucoidan has significant structural diversity: its structure–activity relationship could be investigated to further rational design and modification, achieving efficient clinical development [214].

## 6. Conclusions

In this comprehensive review, we delve into the fascinating properties of natural products sourced from mushrooms, plants, microalgae, and cyanobacteria. These bioactive compounds hold promise in the prevention and treatment of neurodegenerative diseases. Ergothioneine, curcumin, resveratrol, and phycocyanin emerge as powerful allies in the fight against neurodegeneration. Their origins span various natural niches, potentially sharing a common purpose: safeguarding neuronal health. Rather than targeting a single pathway, these compounds adopt multifaceted strategies, and by addressing multiple processes simultaneously, they can mitigate premature cell death and preserve neuronal function. 

While their potential is undeniable, effective delivery to the brain remains a hurdle. Strategies to enhance bioavailability and overcome the blood–brain barrier are critical for translating these discoveries into clinical success. Collaborations between researchers, clinicians, and pharmaceutical industries are essential for the further development of this scientific field. Innovative drug delivery systems hold promise in optimizing the therapeutic impact of these natural neuroprotective agents. Moreover, the gut–brain axis represents a promising target for the development of new therapies for neurodegenerative and neurodevelopmental disorders. Exploiting natural compounds with neuroprotective properties could provide a novel approach to enhance this axis and promote brain health. More research is needed to fully understand the complex interactions between the gut microbiota, the gut–brain axis, and natural neuroprotective niches.

In summary, this manuscript underscores the need to harness the potential of natural compounds for neuroprotection. By bridging the gap between basic research and practical application, the way can be paved for novel treatments, improving the lives of individuals affected by neurodegenerative diseases.

## Figures and Tables

**Figure 1 nutrients-16-01298-f001:**
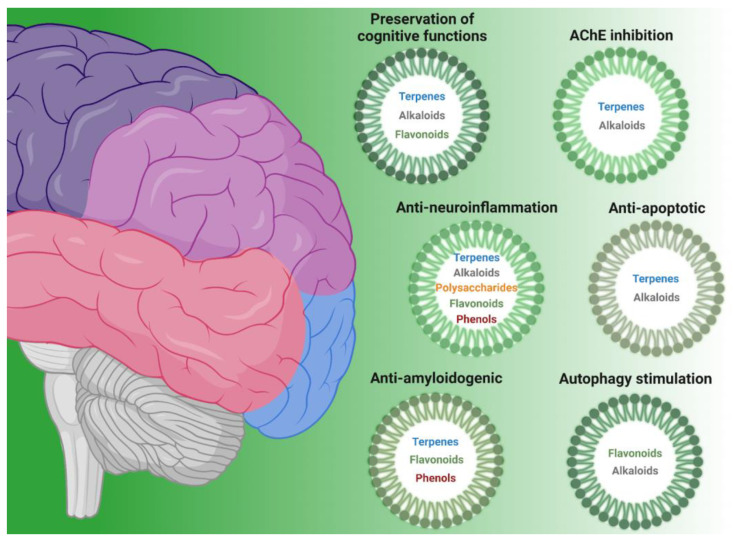
Classes of natural compounds (flavonoids, terpenes, phenols, alkaloids, and polysaccharides) with neuroprotective properties, grouped according to their bioactivity (preservation of cognitive functions, AChE inhibition, anti-neuroinflammation, anti-apoptotic, anti-amyloidogenic, and autophagy stimulation).

**Figure 2 nutrients-16-01298-f002:**
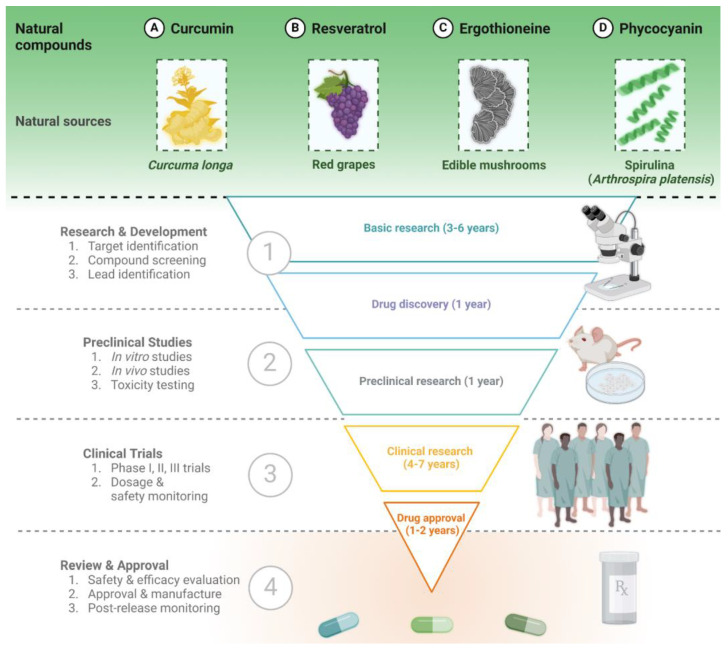
Exemplary natural compounds (curcumin, resveratrol, ergothioneine, and phycocyanin) and the general steps of bioactive substances from natural sources towards clinical approval.

**Table 1 nutrients-16-01298-t001:** Bioactive compounds from natural resources comprehensively classified by the class of bioactivity.

Class of Compound	Compound Name	Natural Source	Ref.
**Anti-amyloidogenic**			
terpenes	Polyozellin, thelephoric acid, polyozellic acid	*Polyozellus multiplex*	[6]
	Erinacine A	*Hericium erinaceus*	[7,8]
	Tanshinone IIA	*Salvia miltiorrhiza*	[9,10]
phenols	Oleuropein aglycone	Extra virgin olive oil	[11]
	Hydroxytyrosol	Extra virgin olive oil	[11]
	Salvianolic acid B	*Salvia miltiorrhiza*	[12,13]
flavonoids	Epigallocatechin gallate	*Camellia sinensis*	[14,15,16]
**AChE inhibitors**			
terpenes	Triterpenes	*Ganoderma lucidum*	[17]
	Polyozellin, thelephoric acid, Polyozellic acid	*Polyozellus multiplex*	[6]
alkaloids	Nostocarboline	*Nostoc 78-12A*	[18]
	Anatoxin-a(s)	*Anabaena flos-aquae strain NRC 525-17*	[19]
	Galanthamine	Amaryllidaceae	[20]
**Anti-apoptotic**			
terpenes	Triterpenes	*Ganoderma lucidum*	[21,22]
	Erinacine A	*Hericium erinaceus*	[8]
	Astragaloside IV	*Astragalus membranaceus*	[23,24]
alkaloids	Caffeine	Coffee and cocoa beans, tea leaves, guarana berries	[25,26]
flavonoids	Fisetin	Strawberry, apple, persimmon, grape, onion, cucumber	[27,28,29,30,31]
	GardeninA	*Gardenia resinifera*, *Tamarix dioica*, *Murraya paniculata*	[32]
**Autophagy stimulation**			
alkaloids	Caffeine	Coffee and cocoa beans, tea leaves, guarana berries	[33]
flavonoids	Isoquercitrin	*Apocynum venetum*	[34,35]
	Hesperidin	*Citrus* fruits	[36]
	Epigallocatechin gallate	*Camellia sinensis*	[37]
	Fisetin	Strawberry, apple, persimmon, grape, onion, cucumber	[38]
	Morin	Mulberry	[39]
**Anti-neuroinflammation**			
polysaccharides	β-glucans	*Grifola frondosa*	[40]
terpenes	Erinacine A	*Hericium erinaceus*	[41]
	Astragaloside IV	*Astragalus membranaceus*	[42,43]
flavonoids	Fisetin	Strawberry, apple, persimmon, grape, onion, cucumber	[44]
	Epigallocatechin gallate	*Camellia sinensis*	[45,46,47]
	GardeninA	*Gardenia resinifera*, *Tamarix dioica*, *Murraya paniculata*	[32]
alkaloids	Caffeine	Coffee and cocoa beans, tea leaves, guarana berries	[33,48,49,50,51,52]
**Preservation of cognitive functions**			
polysaccharides	β-glucans	*Grifola frondosa*	[40]
terpenes	Erinacine A	*Hericium erinaceus*	[7,53,54]
	Astragaloside IV	*Astragalus membranaceus*	[42,55]
alkaloids	Caffeine	Coffee and cocoa beans, tea leaves, guarana berries	[56,57,58,59,60,61]
flavonoids	Epigallocatechin gallate	*Camellia sinensis*	[62,63,64]
	GardeninA	*Gardenia resinifera*, *Tamarix dioica*, *Murraya paniculata*	[32]
	Morin	Mulberry	[65,66]
	Hesperidin	Oranges	[67,68,69,70,71,72]
	Isoquercitrin	*Apocynum venetum*	[73]
phenols	6-Shogaol	Ginger rhizomes (*Zingiber officinale*)	[74,75]
	Hydroxytyrosol	Extra virgin olive oil	[76,77]
	Oleuropein aglycone	Extra virgin olive oil	[11]
	Salvianolic acid B	*Salvia miltiorrhiza*	[78,79]
terpenes	Tanshinone IIA	*Salvia miltiorrhiza*	[80]

## Data Availability

No new data were created or analyzed in this study. Data sharing is not applicable to this article.

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
