# Peer review of "Exploiting Natural Niches with Neuroprotective Properties: A Comprehensive Review"

_nutrients, 2024, doi:10.3390/nu16091298_

Round 1

Reviewer 1 Report

Comments and Suggestions for Authors

In this review, the authors collect the properties of natural products from fungi, plants, microalgae and cyanobacteria, focusing on their potential use in the prevention and treatment of neurodegenerative diseases. In particular, they deal with ergothioneine, curcumin, resveratrol and phycocyanin.

In the final version Table 1 should be condensed, it now takes up many pages.

Author Response

We thank the reviewer for the comments. We changed Table 1, as suggested, halving the number of pages.

Reviewer 2 Report

Comments and Suggestions for Authors

In the present nutrients-2971998 review article, entitled "Exploiting natural niches with neuroprotective properties: a comprehensive review", by Hind Moukham et al, a thorough account is given on the preventive and therapeutic potential of several compounds derived from natural sources, that are exploited for their neuroprotective effect. Curcumin, Resveratrol, Ergothioneine and Phycocyanin are presented herein as examples of successful approaches, with a special focus on possible strategies to improve their delivery to the brain.

In Table 1, a comprehensive list of bioactive compounds from natural resources, classified by their class of bioactivity, is provided.. This is followed by a detailed description of the pharmacological activity of natural drug substances and their therapeutic potential against a variety of serious ailments. These compounds, which are sourced from mushrooms, plants, microalgae, and cyanobacteria, instead of targeting a single pathway, adopt multifaceted strategies, and by addressing multiple processes simultaneously, they can mitigate premature cell death and preserve neuronal function.

The article is concisely written, well documented (228 relevant References are cited), and of interest to both the cognizant and non-cognizant reader.

Author Response

We thank the reviewer for his/her comments and careful reading of the manuscript.

Reviewer 3 Report

Comments and Suggestions for Authors

This manuscript is a good example of e review and gives the scientific community the opportunity to find relevant information regarding natural compounds ant their applicability in the neurodegeneration context.

I suggest that the conclusions section is reviewed. Some of the findings aren't refffered in the manuscript.

Also, it would be an improvement to develop the gut-bain axis section, since it is remarkably important in the Conclusions section.

Authors should consider improving table 1.

Comments on the Quality of English Language

This manuscript is a good example of e review and gives the scientific community the opportunity to find relevant information regarding natural compounds ant their applicability in the neurodegeneration context.

I suggest that the conclusions section is reviewed. Some of the findings aren't refffered in the manuscript.

Also, it would be an improvement to develop the gut-bain axis section, since it is remarkably important in the Conclusions section.

Authors should consider improving table 1.

Author Response

We thank the reviewer for the positive comments.

As suggested, we modified the conclusion, also mentioning the gut brain-axis.

We developed the gut-brain axis, as suggested, in paragraph 4, and we also mentioned it in the Conclusion section.

We agree with the reviewer, and we changed Table 1, as suggested, halving the number of pages.
